

# Performance and applicability of a 2.5D ice-flow model in the vicinity of a dome

Olivier Passalacqua[1], Olivier Gagliardini[1], Frédéric Parrenin[2], Joe Todd[3], Fabien Gillet-Chaulet[2], and Catherine Ritz[2]

[1]Univ. Grenoble Alpes, LGGE, F-38401 Grenoble, France
[2]CNRS, LGGE, F-38041 Grenoble, France
[3]Scott Polar Research Institute, University of Cambridge, Cambridge, UK

*Correspondence to:* Olivier Passalacqua (olivier.passalacqua@univ-grenoble-alpes.fr)

**Abstract.** Three-dimensional ice flow modelling requires a lot of computing resources and observation data, such that 2D simulations are often preferable, at least when the stream lines are parallel; otherwise the lateral divergence of the flow should be accounted for (2.5D models). Assuming that the stream lines follow the steepest slope of the surface, the width variations of a flow tube are computed thanks to the surface curvature. The ability of the 2.5D models to account properly for a 3D state

of strain and stress has not clearly been established, especially their sensitivity on how the ice surface curvature is determined (scanning window on a DEM), and on the geometry of the ice surface. In particular, these models might fail for divergent flows, and need to be more clearly defined. A twin experiment is here carried out, comparing 3D and 2.5D computed velocities, on three dome geometries, for several scanning windows and thermal conditions. The chosen scanning window used to evaluate the ice surface curvature should be comparable to the typical size of the measured ice relief. For isothermal ice, the error made

by the 2.5D model is in the range 0-10 % but for highly diverging flows the errors are 2 or 3 times higher and could lead to a non-physical reversed surface convexity at the dome. For non-isothermal ice, assuming a realistic temperature profile, the presence of a sharp ridge leads to a partly reversed velocity profile. The warmer bottom ice is more deformed than the upper ice, and this results in the non-verticality of the walls of the flow tube, violating the 2.5D assumptions.

## 1 Introduction

Computing performances have been increasing during the last decade, and 3D numerical simulations that could not be performed a few years ago are nowadays affordable. The full-Stokes equations can in particular be solved on large 3D datasets (e.g., Gillet-Chaulet et al., 2012), so that the complete state of stress is accounted for. Nevertheless, two-dimensional $(x, z)$ models are still very useful, since they are easier to handle, the computing time is at least two orders of magnitude less, and they need fewer observations to describe the boundary conditions. Two-dimensional models assume to be unchanging in the

transverse direction, so that they apply well when the topography is always the same in the missing $y$-direction. This can be assumed to be the case for large ice-streams (e.g., Durand et al., 2011), or on the regular flanks of an ice-sheet (e.g., Martín et al., 2006). As a consequence, the ice undergoes a strain in a vertical plane, and no lateral shearing occur (*plane strain* case).





In this study, we call "flow tube" an ensemble of stream lines going through a particular vertical surface (Fig. 1, red lines). If the ice flow is locally converging or diverging, 2D models are no longer valid, as they do not conserve the mass in that case, and cannot account for lateral mechanical stresses imposed by the width variations of the flow tube. It is however possible to account for the width in the model, but still expressing the equations with $(x, z)$ coordinates only; we hereafter call these models

2.5D models. Under a few flow assumptions (flat bed, vertical walls of the flow tube), such models represent an improvement on 2D models because of their better respect of the physics, while maintaining the computational speed which full 3D models lack. A particular case is the circular geometry, where the surface profile is unchanging in any orientation (*axisymmetric* case); this results in the simplest 2.5D model, where the flow tube width has a linear evolution along the flowline.

The 2.5D model proposed by Reeh (1988) assumes that the streamlines are perpendicular to the surface contour lines (Fig. 1),

so that the widening of a flow tube can be described only thanks to the shape of the surface topography. The mass conservation equation then directly depends on the radius of curvature of the surface contour lines (here called $R$). For slowly laterally-varying flowlines and ice domes, the model neglects the horizontal shear stress at the divide. It also implements a vertical profile for the horizontal velocity, by the use of a shape function, and the momentum conservation equation was thus not given for any case. This vertical profile is assumed to be unchanging through the $x$-direction (column-flow model). Later on,

Hvidberg (1993) improved upon this approach by setting the stress equilibrium equations without any assumption on the shape of the velocity field. Furthermore, when considering a width-varying flow tube, the heat equation should be modified as well, but only few authors have considered this as necessary for their purpose (Hvidberg, 1993, 1996; Pattyn, 2002).

Different authors used the modelling approach proposed by Reeh (1988), or similar ones, at different scales and various geometries : for example, mountain glacier (Salamatin et al., 2000; Pattyn, 2002), ice sheets domes (Reeh and Paterson, 1988;

Hvidberg et al., 1997a), ice sheet ridges (Hvidberg et al., 2002) or even at a whole continent scale (Greenland, Reeh et al., 2002). The shape and size of the ice bodies are quite diverse, but unfortunately the validity of the 2.5D approach in these different cases has received little attention. In particular, the principal horizontal strain direction is assumed to be constant along the vertical (i.e. the walls of the tube are vertical), and this has not been properly investigated, whereas it may depend on the surface geometry. Furthermore the error while computing $R$ is only discussed by Hvidberg et al. (1997b), who estimated

the error in the calculation of $R$ by 15 %, and Hvidberg et al. (2001) by about 50 %, but without any decisive argument. The method used to measure $R$ is not detailed either, and we doubt that its influence is negligible.

The determination of the width of the flow tube, or alternatively of the radius $R$, can be done by one of the following means. At a large scale, the ice velocity can be determined thanks to interferometric synthetic-aperture radar data (Rignot et al., 2011), but this could be useful only if the ice velocity is large enough. In the center of ice sheets, where the ice velocity is low,

velocity measurements can be performed thanks to stake monitoring. If correctly positioned, the direction and position of the stakes may give the width of the flow (Waddington et al., 2007). Without any available velocity observations, a digital elevation model (DEM) gives the shape of the surface and its contour lines. The velocity measurement method should be preferred when possible, since the determination of a width is easier and, as we will see, less ambiguous than the one of a curvature. Though, assuming that the surface velocity is oriented along the steepest slope, the surface curvature gives the direction of the flow. To

determine the local curvature of a surface, a DEM is scanned with a computation scanning window, whose width can be freely



chosen, but no objective criteria are currently available to make this choice. This approach was already undertaken to compute the surface curvature of an ice sheet (Rémy and Minster, 1997; Rémy et al., 1999) but they do not discuss the size of their scanning window. Due to the noise of the DEM and to the regional curvature, the local computed curvature can be ambiguous.

No assumptions of the 2.5D model specifically forbid to use it for a highly diverging tube. However, it was already pointed out that the model was not capable to handle such a flow (Reeh, 1989), but this was established for a model assuming a constant horizontal velocity profile. Furthermore, Reeh (1989) accounted for the spatial evolution of $R$ with a simple linear model, based on the surface observation. Similarly, the flow divergence may be a consequence of the basal topography. Sergienko (2012) showed that the 2D $(x, z)$ models could not account for the flow of ice on a bumpy bed, because they consider no lateral variability, and recommended to use these models on a width-averaged topography. These considerations suggest that we do not precisely know for what kind of diverging geometries would the 2.5D model still hold.

As a consequence, the applicability of the 2.5D model should particularly be examined on dome geometries, where simple 2D models would be inoperative and ice flows diverge. The flow tubes may widen by several orders of magnitudes on a few tens of kilometers, especially on the main ridge if the dome is oriented in a preferred direction. The goal of this study is to proceed to several twin experiments for the 2.5D model, for various dome geometries and temperature conditions, and aims at answering the following questions:

1. What is the amplitude of the error made during the process prior to the use of the 2.5D model ? The determination of the radius $R$ leads to a *geometric* error, and this especially depends on the size of the scanning window on the DEM.

2. How well can a 3D state of stress be accounted for by a single 1D parameter along the flow line, depending on the global shape of the topography ? We need to more precisely evaluate the *model* error, consequences of the 2.5D assumptions.

To state the performance of the 2.5D model, we should compare the velocity fields resulting from the 2.5D- and from the 3D model, the latter being taken as a reference. To our knowledge, no such a systematic comparison between the results of 3D and 2.5D models has been carried out, and this work should be done before using these models in different cases, and especially for cases of high divergence. In the following, we first present the equations of the 3D- and the 2.5D model. We then run the results of the simulations on several domes of different shapes: a circular one (axisymmetric), a slightly elongated one, and a very elongated one. The corresponding results allow to build a synthetical DEM, that we use to compute the values of $R$; we then proceed to the 2.5D simulations on the two ridges of the domes (straight flow lines). We first consider the isothermal case, before moving on to investigate the effects of temperature. Finally, we discuss the importance of working with the whole set of the mechanical equations, considering that certain authors used a partial 2.5D model.

## 2 Description of the 3D model

### 2.1 Geometry

Considering that the divergence of the flow might be of critical influence, we work on a ridge of a small dome that will be stretched along a certain orientation. We perform the present model comparison on a similar synthetic geometry which consists of a 15 km-radius domain, whose shape is a quarter of dome only, for reasons of symmetry. The initial thickness of the ice is



3239 m at the dome, the mean surface slope is around 0.6/1000 and the underlying bed is flat. The space coordinate is a $(x, y, z)$ cartesian system.

## 2.2 Mesh

The 3D mesh is horizontally unstructured and vertically extruded on 10 levels. The horizontal mean spacing between the nodes is 1 km (Fig. 2).

## 2.3 Mechanical model

### 2.3.1 Conservation equations

We call $\boldsymbol{u}$ the velocity vector of components $(u, v, w)^t$. The stress and strain rate tensors are denoted $\boldsymbol{\sigma}$ and $\boldsymbol{\epsilon}$ respectively, and their components, $\sigma_{ij}$ and $\epsilon_{ij}$. The deviatoric part of $\boldsymbol{\sigma}$ is denoted $\boldsymbol{\tau}$, and its components, $\tau_{ij}$. The 3D mechanical model consists of a Stokes problem for incompressible ice of density $\rho$, in which the mass and momentum conservations equations are written

$$\nabla \cdot \boldsymbol{u} = 0 \tag{1}$$

$$\nabla \cdot \boldsymbol{\sigma} + \rho \boldsymbol{g} = 0 \tag{2}$$

where $\boldsymbol{g}$ is the gravitational acceleration vector. The values of the different parameters are given in Table 1. The ice is assumed to deform following Glen's generalized flow law (Glen, 1958):

$$\dot{\epsilon}_{ij} = A(T) \tau_e^{n-1} \tau_{ij} \tag{3}$$

where $\tau_e$ is the second invariant of $\boldsymbol{\tau}$. We choose a value of $n = 3$. The rate factor $A(T)$ non linearly depends on temperature, reflecting the fact that warm ice is softer. To study the influence of the temperature $T$ (expressed in kelvin) on the performance of our 2.5D model, we first consider isothermal ice at $245\,\mathrm{K}$, and then a non-isothermal ice, for which $T(z) = 270 - 50 \cdot (z - b)/(s - b)$, where $s$ and $b$ are the altitudes of the surface and the bedrock, respectively. This linear temperature profile is simple but realistic enough for our purpose. For convenience, and as the bed is flat, $b = 0$ in the following experiments.

### 2.3.2 Boundary conditions

Since the 3D mesh is a quarter of a dome, the conditions have to be set on 5 different boundaries, numbered from BC1 to BC5 (Fig. 2). We consider a frozen ice at the bed and no output flow on the lateral boundaries (symmetry conditions). Considering no sliding at the bottom and neglecting the atmospheric pressure at the surface, the boundary conditions are as follows:





$$BC1 : \boldsymbol{u}.\boldsymbol{n}|_{y=0} = 0 \tag{4}$$

$$BC2 : \boldsymbol{u}.\boldsymbol{n}|_{x=0} = 0 \tag{5}$$

$$BC3 : \boldsymbol{u}|_{z=0} = \boldsymbol{0} \tag{6}$$

$$BC4 : \boldsymbol{\sigma}.\boldsymbol{n}|_{z=s} = 0 \tag{7}$$

where $\boldsymbol{n}$ is the vector pointing outward the surface. Since the surface (BC4) is let free, a kinematic boundary condition for the surface $s(x,y,t)$ has to be solved as well:

$$\frac{\partial s}{\partial t} + u\frac{\partial s}{\partial x} + v\frac{\partial s}{\partial y} = w + a \tag{8}$$

where $a$ is the accumulation function. The boundary conditions BC5 are of particular interest since they are set to control the shape of the steady-state dome. The method used to create an elongated dome is similar to that of Gillet-Chaulet and Hindmarsh (2011): the shallow ice approximation is used to prescribe a profile of horizontal velocity on BC5, and the output velocities are tuned depending on their orientation, so that the shape of the dome is elongated in one preferred direction (Fig. 3):

$$u = \overline{\omega}\frac{2\cos\theta}{\alpha}\frac{(n+2)}{(n+1)} \times \left(1 - \left(1 - \frac{z}{s}\right)^{(n+1)}\right) \tag{9}$$

$$v = \overline{\omega}\frac{2(\alpha-1)\sin\theta}{\alpha}\frac{(n+2)}{(n+1)} \times \left(1 - \left(1 - \frac{z}{s}\right)^{(n+1)}\right) \tag{10}$$

where $\theta$ is the angle to the edge of the domain, and $\alpha$ a shape parameter controlling the elongation of the dome. The following results will correspond to a stabilized steady-state geometry, for a constant accumulation $a$ in time and space. To do so, the output mean velocity $\overline{\omega}$ is tuned to balance the surface accumulation, so that it can be expressed as

$$\overline{\omega} = \frac{a \cdot \Sigma}{W_L \cdot s} \tag{11}$$

where $\Sigma$ is the upstream feeding area on the top boundary (gray area in Fig. 1), and $W_L$ the width of the ice flow at the downstream position. In this particular case, $\Sigma$ and $W_L$ are simply equal to 1/4 of the dome surface and dome perimeter, and $L$ is the radius of the dome. Three different values were taken for the shape parameter $\alpha$: 2 (axisymmetric case, circular geometry), 3, and 6 (Fig. 3). The first one will inform on the performance of the model for a perfectly known case, and the ability to estimate the value of $R$. Then the other ones will inform on the capability to account for a more or less divergent flow, and the caveats to be aware of.



## 3  Description of the 2.5 D model

The coordinate system used by Reeh (1988) and Hvidberg (1993) is a curvilinear coordinate system with right-handed oriented coordinate axis. The $x$-axis is oriented along the flow line, the $z$-axis is vertically oriented, and the $y$-axis is transverse to flow, and tangential to a surface contour line. As we only consider here straight flow lines (linear ridge of an ice divide), the coordinate system is locally cartesian (Fig. 1). We refer to the 2.5D model in the $(x, z)$ coordinate system. We now recall the assumptions made by Hvidberg (1993), partly inherited from Reeh (1988).

1. The flowlines are perpendicular to the surface contour lines.

2. The direction of the horizontal velocity components are constant with depth, that means that we assume the walls of the flow tube to be vertical.

3. There is no shear stresses on the vertical boundaries defined by the flow tube.

4. The ice deforms according to Glen's flow law.

These assumptions together mean that the surface geometrical strain is transferred to the bottom, so that the surface contour lines and the horizontal velocity in the flow direction impose the transverse stresses. Such assumptions are reasonable in the center of an ice-sheet for a slowly varying bed. If the bedrock spatial variations are too steep, they will warp the ice free surface so that the bottom flow may not be parallel to the surface flow (Hvidberg, 1993; Sergienko, 2012).

### 3.1  Geometry

The 2D domain is taken as a vertical slice of the 3D domain, on one of its lateral boundaries (Fig. 2). The dome being elongated, we will run the model along the sharpest ridge ($y = 0$) or perpendicular to it ($x = 0$).

### 3.2  Mechanical model

We call $W(x)$ the width of the considered flow tube. The radius of the surface contours lines $R(x)$ is taken positive for diverging flow and negative for converging flow. In this model, the assumption (2) implies that $W$ has no dependence on $z$, so that geometrical considerations show that $W(x)$ is directly linked to $R(x)$ by

$$\frac{1}{R(x)} = \frac{1}{W(x)} \frac{\partial W}{\partial x} \tag{12}$$

An axisymmetric dome leads to the simple relations $R(x) = x$ and $W(x) \propto x$. If the flow tube is diverging more than the axisymmetric flow (on a ridge for example), the corresponding tube surface is narrowed for a given output width, and leads to lower output velocities.

The following sections present the equations of mass and momentum conservation, modified to account for the divergence of the tube. As the velocity field mainly depends on the input/output balance, some authors only conserve the mass (e.g.,



Parrenin et al., 2004; Todd and Christoffersen, 2014), but not the momentum; this approach can in particular be sufficient for a vertically-integrated model (Hvidberg et al., 1997b). Other authors do not even modify the mass conservation equation, but add an extra-surface mass balance term (Cook et al., 2014; Gladstone et al., 2012) which depends on the divergence of the tube. This approach has the advantage of simplicity and results in a correct output flux, but neglects the true horizontal advection of

the ice. However, this can be justified for ice sheet margins, where the ice mainly undergoes sliding. For all these mass-only conservation models, the normal lateral stress of the surrounding ice is not accounted for, since the force equilibrium is not properly modified.

### 3.2.1 Mass conservation

As we stand on an ice divide, the velocity component $v$ and its spatial derivative vanish for reason of symmetry, so that there

is no dependence of the strain rates on the transversal coordinate. Under the above assumptions and considering a flow tube of width $W(x)$ (and corresponding radius $R(x)$), the normal strain rates in the curvilinear system are then made simple (Jaeger, 1969):

$$\dot{\epsilon}_{xx} = \frac{\partial u}{\partial x} \, ; \, \dot{\epsilon}_{yy} = \frac{u}{R(x)} \, ; \, \dot{\epsilon}_{zz} = \frac{\partial w}{\partial z} \tag{13}$$

If the flow tube has a constant width, the value of $R$ is infinite and the equation correspond to a plane strain case. For the

more complete form of these expressions, see the discussion of Reeh (1988). The mass conservation then follows:

$$\frac{\partial u}{\partial x} + \frac{u}{R(x)} + \frac{\partial w}{\partial z} = 0 \tag{14}$$

### 3.2.2 Momentum conservation

No specific assumption is made on the velocity profile. The force equilibrium equations, expressed in the $(x, z)$ coordinate system, is thus written (Jaeger, 1969; Hvidberg, 1993):

$$\frac{\partial \sigma_{xx}}{\partial x} + \frac{\partial \sigma_{xz}}{\partial z} + \frac{\sigma_{xx} - \sigma_{yy}}{R(x)} = 0 \tag{15}$$

$$\frac{\partial \sigma_{xz}}{\partial x} + \frac{\partial \sigma_{zz}}{\partial z} + \frac{\sigma_{xz}}{R(x)} = \rho g \tag{16}$$

where $\sigma_{yy}$ is known in terms of $u$ and $R(x)$ thanks to Eqs. (3) and (13).





### 3.2.3 Boundary conditions

The boundary conditions are inherited from the 3D case: no sliding, free surface, vanishing velocity upstream of the domain and an imposed vertical velocity profile downstream. Note that the value of the mean output velocity $\overline{\omega}$ directly depends on $\Sigma$, which is now given by

$$\Sigma = \int\limits_0^L W(x)dx \qquad (17)$$

where $L$ is now the length of the flow line. Equations (11) and (17) together mean that the errors in the calculation of $W$ result in errors in the prescribed output velocity. Since we stand on a straight flow line, there is no transverse flow across the considered plane. The free surface equation is thus derived from Eq. (8), and is equivalent to a simple 2D case, here given for the $(y = 0)$ plane:

$$\frac{\partial s}{\partial t} + u\frac{\partial s}{\partial x} = w + a \qquad (18)$$

### 3.3 Implementation in Elmer/Ice

The modified mechanical equations is implemented in the Elmer/Ice finite element software (Gagliardini et al., 2013), which is known to efficiently account for the set of full-Stokes equations. The correct implementation of the mass conservation was checked by comparing different 2.5D simulations with the Vialov-type profiles (Vialov, 1958) computed for different diverging tubes. The expression of the Vialov profile in the case of a power-law varying flow tube is presented in Appendix A.

### 3.4 Determination of the contour radius

To determine the radius of curvature of the surface contour lines, we first export a DEM from the surface nodes of the 3D model. These nodes are interpolated using an inverse distance weighting, with a power of 4 in order to ensure a good smoothing of the computed surface, representative of a real ice sheet. To be close to a real case, the spatial resolution is taken equal to $400\,\mathrm{m}$, which is the resolution of the DEM resulting from the ICESat mission on Antarctica (Schutz et al., 2005). Then, the method consists of selecting the width of a scanning window (Fig. 1), which is used to fit the surface of the DEM by a bivariate quadratic polynomial function. Three different sizes of scanning window were tested: $2.8\,\mathrm{km}$, $6\,\mathrm{km}$ and $10\,\mathrm{km}$, thus corresponding to a width of 7, 15 and 25 pixels.

### 3.5 Protocol of comparison

We first run the 3D transient isothermal simulation, and stop when a steady state is reached ($\partial s/\partial t < 10^{-6}\,\mathrm{m/a}$). We then use the resulting DEM to compute the profile of $R$ to initialize the 2.5D model.





For each dome configuration ($\alpha = 2,3,6$) we compare the different runs chosen: a) 3D b) 2.5D for the three scanning windows, with a fixed geometry c) 2.5D for the three scanning windows, with a free surface. For the axi-symmetric case we add a true 2D axisymmetric run (imposing $R = x$).

Then we compare the 3D and 2.5D results for a variable temperature ice, to see the influence of the temperature, especially
near the base of the ice sheet. We finally compare the results between the velocity field output from this 2.5D model and from the same model conserving the mass only, thus neglecting equations (15) and (16).

## 4   Results and discussion

### 4.1   Circular geometry ($\alpha = 2$), isothermal ice

#### 4.1.1   3D/2D axisymmetric comparison

The absolute error on the ice velocity for an axisymmetric 2D ($R = x$) model is of the order of $10^{-4}\,\mathrm{m/a}$ (Fig. 4.a, where the black and yellow curves are almost superimposed). The free surface case shows almost no change in surface elevation, and the result is very close to the fixed geometry case. The observed error is a result of discretization, and should tend to zero as the element sizes decrease.

#### 4.1.2   3D/2.5D comparison

The computation of the radius of the surface contour lines is strongly influenced by the size of the scanning window. For a circular geometry, the variation of $W$ along $x$ should be linear, which is almost the case for the two wider windows (Fig. 5). With this regular geometry, the larger the window, the more precise the radius, since the interpolation surface will be more accurate. On the contrary, the width value computed with the smaller window is less regular and underestimated by about 30%, meaning that we cannot evaluate a certain curvature from too small a sample. If choosing an appropriate scanning-window size
(i.e. not too small), the range of the error is about 0-10 % (Fig. 4.a), which is a consequence of the error in the calculation of the radius.

### 4.2   Elongated domes ($\alpha = 3$ and $\alpha = 6$), isothermal ice

For elongated domes, we consider both the flow line along the sharpest ridge ($y = 0$), and perpendicular to it ($x = 0$).

#### 4.2.1   3D/2.5D comparison, along the ridge

Along a ridge, a flow tube is non linearly diverging. For a given output width, the accumulation area is smaller than in the axisymmetric case, thus leading to lower output velocities.

With fixed geometries, it clearly appears that the velocity is underestimated for elongated domes (Fig. 4.b and 4.c, dashed lines), meaning that the local surface slope cannot explain the ice motion by itself: the ice along the ridge is also, if not mainly,



pulled by the surrounding lateral ice, which moves thanks to a steeper surface slope. The case of the small scanning window appears to be different (Fig. 4.b, red dashed line), simply because of a really bad estimation of $W$, thus of the output velocity.

The downstream velocities are always quite accurate (10 % of error), since they mainly depend on the tube surface calculation, incorporated in the velocity boundary condition. When releasing the surface, the surface slope slightly increases to

accomodate the velocity boundary condition, and the computed velocity field is then closer to the 3D reference. The relative error made in the downstream part of the flow is comparatively higher near the divide since the velocities are very small.

A sharp ridge leads to a high divergence of the flow. In particular, the 2.5D model seems to fail reproducing the 3D flow for $\alpha = 6$, because the flow divergence cannot be accomodated by the surface near the dome unless the surface convexity is inverted (its slope decreases with increasing $x$, Fig. 6). Since the vertical strain rate is always of the order of $a/H$, the

conservation equation leads to a balance between $\partial u/\partial x$ and $u/R$. Simple considerations (Appendix B) show that $u/R$ should be very small and the horizontal strain rate high, so that the surface slope has to be steeper. To handle this artefact, we tried to increase the mesh resolution near the dome, but without any successful results. This artefact does not appear for $\alpha = 3$ (slightly elongated dome).

For $\alpha = 6$, the tube surface is better estimated with an intermediate window, whose size is closer to the local value of $R$.

Too large a window would consider the whole shape of the dome and lead to an underestimation of $R$. The amplitude of the error between the different runs show that for sharp ridges (or highly diverging tubes) the choice of the window size is not straightforward, as a wider window nevertheless increases the regularity of the velocity field.

Reeh (1989) explained the unsatisfying results for the diverging tube of Camp Century partly by the simplicity of his model, especially his linear model of $R$. Indeed, the error made with this more complete model seems now to be small enough to be

used for dating purposes (for example with the intermediate window size). However, for a real case of highly curved surface, it is difficult to know *a priori* the best window to use and we may stand out of the applicability domain of the model.

### 4.2.2   3D/2.5D comparison, perpendicular to the ridge

For the flow line perpendicular to the sharpest ridge, the error is of the same order of magnitude – or a little bit higher – than for the circular geometry (Fig. 4.d), and for the two wider windows only if the flow is highly divergent (Fig 4.e). In the case

of large radius values, towards the exterior of the dome, the wider window gives the more accurate results. The error is here consistent with the estimation of Hvidberg et al. (1997b), who considered a similar flow line between the two drilling sites of GRIP and GISP.

### 4.3   Non-isothermal ice

A supplementary comparison is lead on the sharp ridge ($y = 0$) for a temperature varying linearly through depth. The computed

velocity field towards the divide (low $x$ values) shows a reversed vertical profile, i.e. the bottom ice goes faster than the upper ice (Fig. 7, bottom). This non-physical result in 2.5D can be explained as follows: as soon as a 3D tube diverges more than for an axisymmetric flow, the warmer bottom ice is more easily laterally strained than the colder surface ice. As a consequence, the walls of the 3D flow tube are not vertical anymore, and using the 2.5D model in such a case would violate the assumption (2).





This effect is particularly pronounced close to the divide, where the tube is narrower, and can be seen in the 3D velocity field. The stream lines going through a vertical line at $x = 1000\,\mathrm{m}$ diverge more or less, a few hundred of meter further, depending on their depth (Fig. 8, blue curves), and this dependence is stronger for high diverging tubes. To accomodate the lateral strain in this case, the 2.5D model computes high horizontal velocities in the bottom, whereas the real motion is in fact mainly laterally

oriented. No mesh refinement has been able to correct this problem. Since it does not happen for a constant temperature, it certainly originates from the lower viscosity of the bottom ice (Fig. 8, red curves).

This result also suggests that, on sharp ridges and with non-isothermal ice, working with a fixed vertical profile of velocity will prevent from such unintended behaviour. It does not invalidate the conclusions of the model of Hvidberg et al. (2002) so far, since their ice ridge was sharpest downstream than upstream, which is opposite of what is done here, and their flowline was

much longer. Nevertheless, care must be taken in such cases, since the basic assumptions may not be respected and the model is likely to be outwith its application domain.

For reasons of continuity, the walls of the flow tube should not be vertical in the direction perpendicular to the ridge as well, but the effect is too weak to impact the computed velocity field.

### 4.4 Influence of the momentum conservation

The velocity fields computed by the complete 2.5D model and the mass-only conservation model for the axisymmetric case show no substantial difference (Fig. 9). The highest discrepancy is of the order of 2.5 % of the velocity value. This result is quite surprising since we could have expected a much more sensitive influence of the conservation of momentum in such a diverging flow, or difficulties to account for the divergence when neglecting it. As the ice is an incompressible medium, the velocity field seems in this particular case to be mainly determined by the mass conservation only. However, it does not mean

that the force equilibrium should be neglected in any case, since our simulations are performed for a synthetic case (flat bed, linear flow line, symmetric ridge, and steady state). Furthermore, the present domain is relatively small; for larger domains, working with the complete set of 2.5D equations should be more conservative.

## 5 Conclusions

A systematic comparison between 2.5D- and 3D models has been presented in order to evaluate the ability of the former to

accurately compute the velocity field on a small dome of an ice sheet. The error made when estimating the value of the radius of the surface contour lines is of the order of 10% if the computation window is well chosen, though it can be comparatively higher close to the divide. The radius of curvature of the surface elevation contour lines should be determined with a sufficiently large computation window, but choosing the optimum width is not completely straightforward; in any case we suggest it should not be less than one-third of the maximum measured radius, and several windows should be tested to ensure the robustness of

the results.

The 2.5D model can be used without any specific restriction for tubes diverging less than and up to an axisymmetric flow. For isothermal ice, the model can be used with tubes diverging more than an axisymmetric flow, if the divergence is not too





high. For very high divergence, the ice is in fact mainly pulled by the output boundary condition, and the resulted velocity field and surface geometry may be somewhat irregular.

For non isothermal ice, the tube should not diverge more than axisymmetry, because the softer bottom ice would be much easily laterally strained in the case of an elongated dome. The walls of the flow tube are therefore not vertical, which violates the model assumptions, and the corresponding horizontal velocity profile may be not physical near the divide.

**Code availability**

The presented simulations were performed thanks to the finite element model Elmer/Ice v.7.0 rev. 7016. The source code of the 2.5D model is available in the distribution since v.8.0 rev. d9d4a2f, implemented in the AIFlow solver.

**Author contribution**

The experiments were designed by OG and FP. OP carried them out, helped by CR for analytical developments, and by FGC for the Elmer/Ice implementation. OP and JT proceeded to the comparison between complete and partial 2.5D models. OP prepared the manuscript with contributions from all co-authors.

**Appendix A: Vialov profile for a power-law diverging tube**

To check the correct implementation of the mass conservation in the 2.5D model, we hereafter compute the height of a Vialov profile corresponding to a regularly diverging flow tube. Note that such a surface is only representative of what happen for a single flow line, and not for a whole surface, as what is usually done for a Vialov profile in plane strain (Vialov, 1958) or axisymmetry (Ritz, 1992).

Figure 5 shows that we may approximate the shape of the flow tube by a power-law depending on the $x$-coordinate. Let consider a flow tube of width $W = W_L \left( \frac{x}{L} \right)^{\beta}$. For plane flow, $\beta = 0$, for axisymmetry $\beta = 1$, and for sharp ridges $\beta > 1$. The volume outflow $q^*$ for a certain coordinate $x$ may be expressed in one of two ways (Cuffey and Paterson, 2010, p.388):

$$q^* = \int_0^x a \cdot \left( \frac{x'}{L} \right)^{\beta} W_L \, dx' = \frac{a \cdot x}{(\beta + 1)} \cdot \left( \frac{x}{L} \right)^{\beta} W_L = \frac{a \cdot x \cdot W}{(\beta + 1)} \tag{A1}$$

$$q^* = \left( \frac{2A}{n+2} \tau_b^n H \right) HW \tag{A2}$$

where $H$ is the ice thickness, $\tau_b$ the basal shear, $a$ the accumulation rate, $L$ the length of the glacier, $A$ the rate factor of Glen's flow law. Equating the two expressions yields

$$a \cdot x = \frac{2 \cdot (\beta + 1) \cdot A}{n+2} \tau_b^n H^2 \tag{A3}$$





The following reasoning is then similar to the one of Cuffey and Paterson (2010), simply modified by a $(\beta + 1)$ multiplier. The final expression for the ice thickness is unchanged

$$H = H^* \left( 1 - \left( \frac{x}{L} \right)^{\frac{n+1}{n}} \right)^{\frac{n}{2n+2}} \tag{A4}$$

except the height of the ice sheet $H^*$ at the dome $(x = 0)$, which is now

$$5 \quad H^* = \left( \frac{2(n+2)^{1/n}}{\rho g} \right)^{\frac{n}{2n+2}} \sqrt{L} \left( \frac{a}{2 \cdot (\beta + 1) \cdot A} \right)^{\frac{1}{2n+2}} \tag{A5}$$

This expression is consistent with the one already derived for axisymmetry. We then use this expression to control the 2.5D model by comparing the value of $H^*$ computed by the model with its above theoretical value.

## Appendix B: Radius and surface of a power-law diverging flow tube

We consider the same flow as in Appendix A. The value of the radius $R(x)$ is then expressed as

$$10 \quad \frac{1}{R(x)} = \frac{1}{W} \frac{dW}{dx} = \frac{dln(W)}{dx} = \frac{\beta}{x} \tag{B1}$$

The surface area of the tube upstream of $x$ can be expressed as

$$\Sigma(x) = \int_0^x W(x)dx = \frac{W_L}{\beta + 1} \frac{x^{\beta+1}}{L^\beta} \tag{B2}$$

As $u$ is more or less proportionnal to the upstream surface area $\Sigma$, $u/R$ is expected to be proportionnal to $W_L \left( \frac{x}{L} \right)^\beta$. On the contrary, one can consider that the value of $\partial u/\partial x$ should be of the order of $\overline{\omega}/L$, i.e. simply proportionnal to $\Sigma(L)/L = W_L/(\beta + 1)$. Near the divide, $u/R$ is then comparatively much smaller than $\partial u/\partial x$ for sharp ridges than for axisymmetric flows, and imbalances the corresponding mass conservation.





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





**Table 1.** Description and values of the model parameters.

| Parameter | Variable | Value | Unit |
|---|---|---|---|
| Accumulation rate | $a$ | 0.04 | m/a |
| Flow line length | $L$ | 15 000 | m |
| Downstream width | $W_L$ | 1 | m |
| Flow law exponent | $n$ | 3.0 | |
| Initial max. ice thickness | $\max(s)$ | 3239 | m |
| Ice density | $\rho$ | 917 | kg/m$^3$ |
| Mean surface slope | | 0.6 / 1000 | |

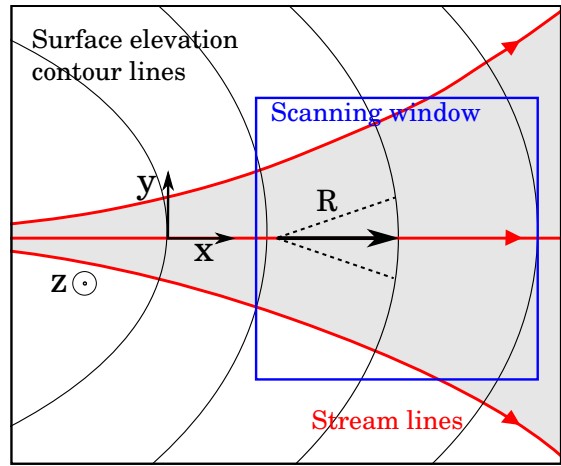

**Figure 1.** The $x$-axis is taken along the ridge stream line, and the $y$-axis tangent to the surface elevation contour. The scanning window is used to determine the value of $R$ in its center. Lateral stream lines are represented to show the non linear widening of the flow tube in this particular case, and the corresponding accumulation surface is colored in gray.





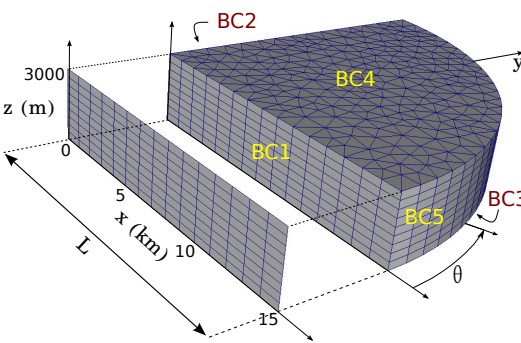

**Figure 2.** View of the two meshes used in this study. For each run, the 2D mesh is extracted from the geometry of the steady state solution of the 3D simulation. BC1 to BC5 refer to the 5 boundary conditions of the 3D case.

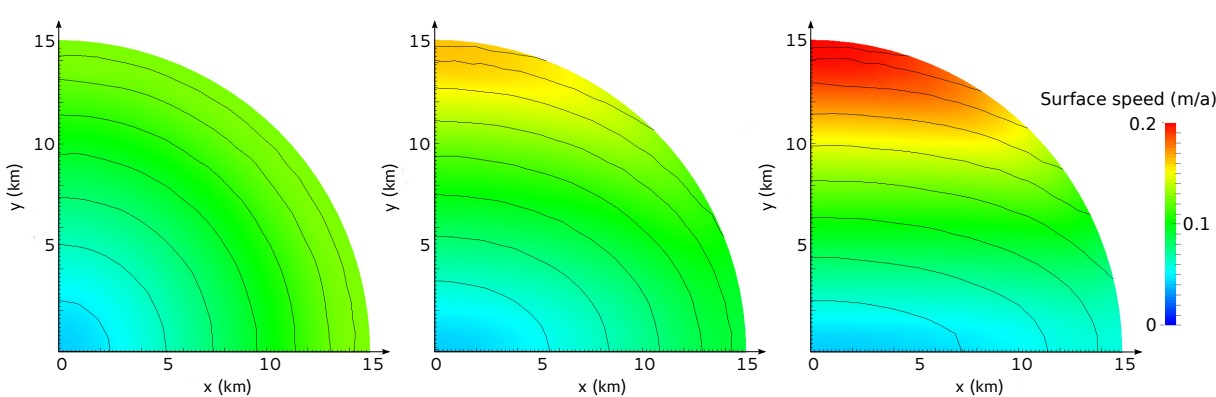

**Figure 3.** The three domes stabilized at steady state: surface contour lines (spacing: 1 m), and surface speed (m/a) for (a) $\alpha = 2$, (b) $\alpha = 3$ and (c) $\alpha = 6$.



**Figure 4.** Horizontal velocity at the ice surface (m/a) along the flow line, for isothermal ice. Dashed lines are for fixed geometry runs, and solid lines for evolved free surface to a steady state. Thick black : 3D; yellow: 2D axisymmetric (hidden by the black curve); red: 2.5D scanning window of 2.8 km; green: 2.5D scanning window of 6 km; blue: 2.5D scanning window of 10 km. a) Circular dome, $\alpha = 2$. For readability reasons, the fixed geometry runs are not shown here. b) $\alpha = 3$, along the ridge. c) $\alpha = 6$, along the ridge. d) $\alpha = 3$, perpendicular to the ridge. e) $\alpha = 6$, perpendicular to the ridge. For this later simulation, no result concerning the 2.8 km scanning window is shown since they are completely out of reasonable bounds.




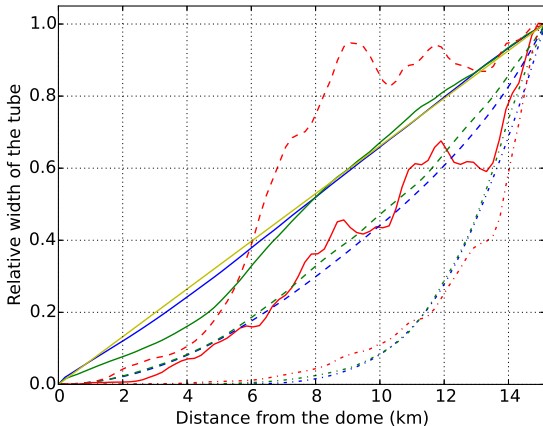

**Figure 5.** Relative width of the flow tube $W$ for the different dome geometries: circular ($\alpha = 2$, solid lines), slightly elongated ($\alpha = 3$, dashed lines) and very elongated ($\alpha = 6$, dotted lines); for different scanning windows: $2.8\,\mathrm{km}$ (red), $6\,\mathrm{km}$ (green) and $10\,\mathrm{km}$ (blue). The yellow curve is the line for which $R = x$ is imposed (axisymmetric case).

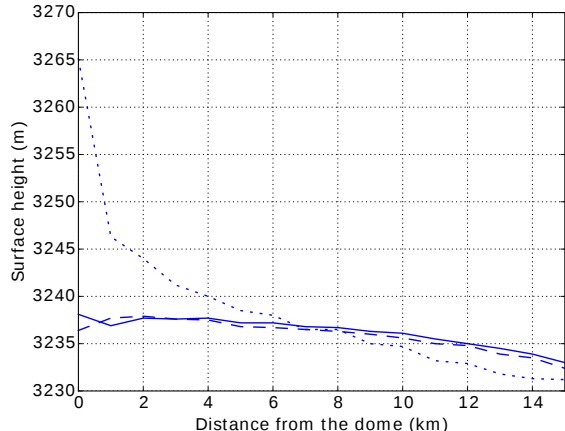

**Figure 6.** Height of the ice surface (m) for the 2.5D free surface model, along the ridge. Solid line: circular geometry, $\alpha = 2$. Dashed line: $\alpha = 3$. Dotted line: $\alpha = 6$.





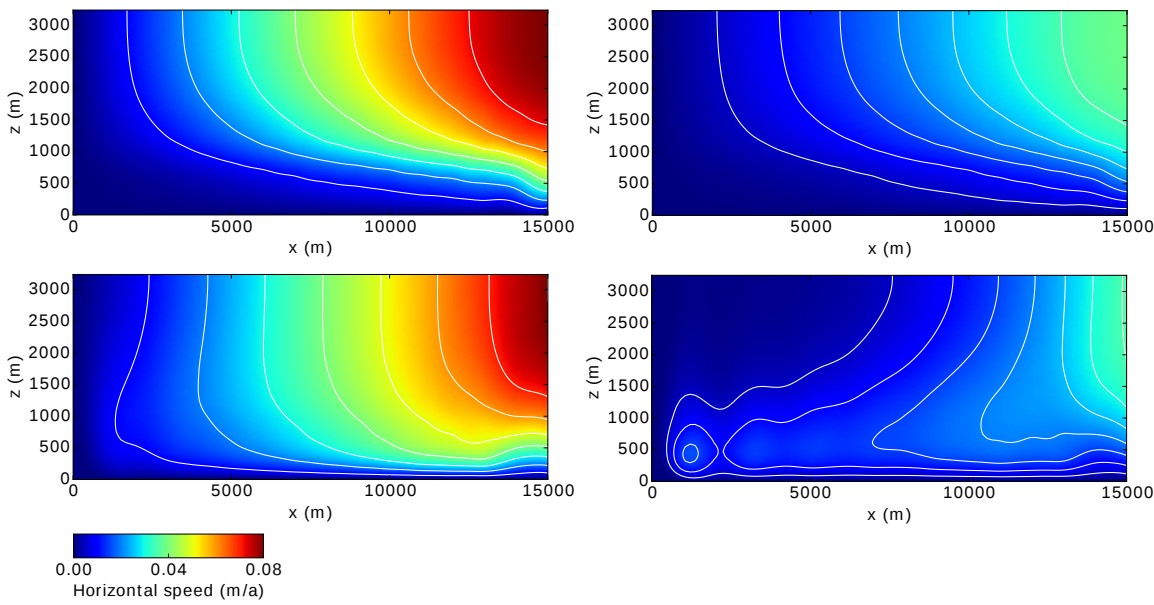

**Figure 7.** Horizontal velocity field (m/a), for the non isothermal case and $\alpha = 3$. Top: 3D model. Bottom: 2.5D model, with a scanning window of 10 km. Left: $\alpha = 3$, white contour lines spaced by 0.01 m/a. Right: $\alpha = 6$, white contour lines spaced by 0.005 m/a.

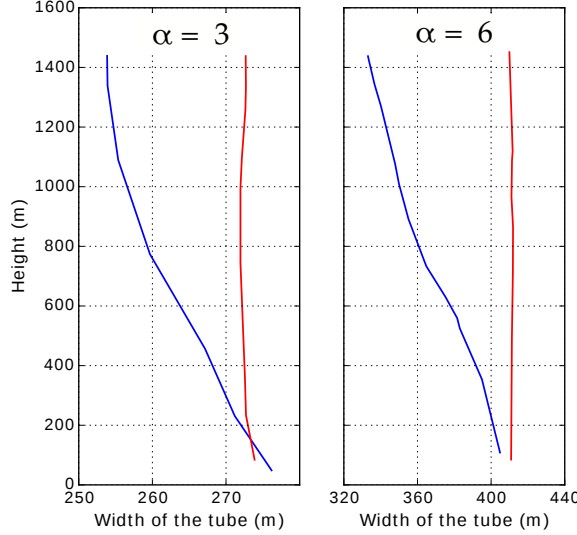

**Figure 8.** Width of the flow tube (m), for the isothermal (red) and non isothermal case (blue), for $\alpha = 3$ ($x = 1500$ m, left) and $\alpha = 6$ ($x = 1200$ m, right) The aspect of the curves is slightly affected by numerical noise due to discretization.





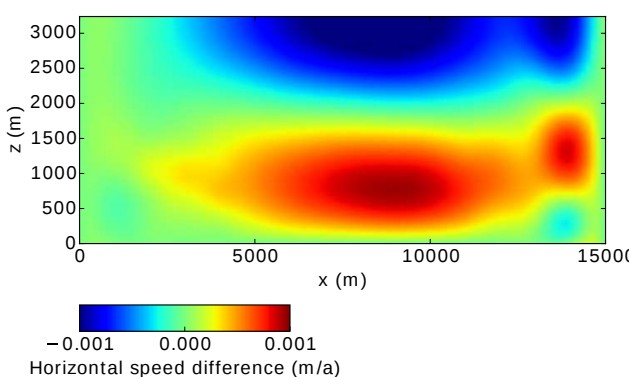

**Figure 9.** Horizontal speed difference between the mass-only conservation model and the complete 2.5D model (m/a), for $\alpha = 2$ and isothermal ice.