# Peer review of "Performance and applicability of a 2.5D ice-flow model in the vicinity of a dome"

_Geoscientific Model Development, 2016_

## Referee Comment (RC1) · D. Brinkerhoff (Referee) · 17 Feb 2016

In *Performance and applicability of a 2.5D ice-flow model in the vicinity of a dome* the authors compare the flow velocities (and, to a lesser extent, geometry) produced by the Stokes' solver Elmer/Ice to that produced by a so-called 2.5D model, which uses a paramaterization of flowline width to reduce the dimensionality of the model equations while still approximating the full 3D physics. A critical parameter in this process is the flowline width, which they approximate from surface elevations using an ad-hoc (but largely unspecified) method. They find that the 2.5D model works very well when the width is specified exactly, and that much of the error in the 2.5D model output is a result of inaccuracies in flowline width, which can be exacerbated by using an incorrect sample size when evaluating surface elevations. Additional complications arise when conditions are non-isothermal.
[Figure]

General Comments:

For this paper to warrant publication, it would require its methods to be outlined in much greater detail than they currently are. The primary nuance of the work, mainly the computation of $R$ from a surface DEM is given only a few lines of description. The metrics by which the authors judge quality are primarily left unstated. Many modelling assumption (i.e. that surface elevations can act as a reasonable proxy for the direction of a Stokes'-based velocity field) are utilized but unjustifed. Suffice to say, this work would not be repeatable based upon the information given here. I would like to offer up an opinion on whether the results presented are significant to the field of glaciology, but lack a sufficient understanding of the quality of the results to do so.

The paper is rife with incorrect grammar, spelling, and otherwise improper English, to the extent that it partially inhibits understanding of the methods and results contained herein. As a reviewer, I don't really see it as my responsibility to correct for these issues, particularly when at least one of the co-authors is a native English speaker, and I will not provide comments on these points. As such, I would suggest that the authors employ or otherwise recruit a capable copy editor to polish the manuscript. It is presently not ready for publication for this reason, scientific merit aside.

Specific Comments:

Abstract: the abstract introduces a significant amount of jargon, such as "scanning window", "reversed surface convexity", and "partly reversed velocity profile", which the reader cannot know the meaning of without reading the paper. This undercuts the purpose of the abstract.

P1 L8: The relief in question is on the order of tens of meters, yet the authors suggest that the so-called "scanning window" should be on the order of kilometers. This directly contradicts this line.

P1 L16: the equations in question are either Stokes' equations, or they are not; the

"full-" modifier is unnecessary.

P1 L21: Durand (2011) does not appear to actually address this issue, though it does use a flowline model. Also, what are the "regular flanks of an ice sheet"?

P1 L22: "plane strain" does not seem to be used correctly here.

P2 L1: I think I know what is meant by "a particular vertical surface", but this would be greatly clarified by addition of said surface to Fig 1.

P2 L14: Why is the parenthetical "(column-flow model)" included here? It does not seem to serve a purpose and doesn't seem to be referenced later.

P2 L22: "In particular,..." I do not understand this sentence.

P2 L31: width of the flow *tube*?

P2 L31: A DEM gives the shape of the surface and its contour lines when velocity is available, as well.

P2 L34: Assuming that flow is always oriented along the surface gradient is a wrong assumption, and the differences between considering gradient-based curvature and velocity-based width goes far beyond issues of numerical accuracy. Have a look at any computation of balance velocity ever made; if flow is routed down-gradient then the surface velocity looks like a river network, which is silly. Longitudinal stresses matter for flow routing!

P2 L35: I don't know what "scanned" means, and as such I cannot assess whether the authors' statements about ambiguity in surface curvature have any merit. It would be better to hold off on elaborating upon it further until some basic definitions have been stated.

P3 L10: On "diverging geometries": flow fields can diverge, because they are vector fields. Scalars cannot diverge, and the fields representing geometry are certainly scalars.
P3 L17: What "1D parameter" are we talking about here? Where is this question returned to in the rest of the text?

P3 L25: Please state more clearly that the 3D model is used to construct a set of surface geometries based on different choices of lateral boundary conditions. This 3D model output is then taken as input to the 2.5D model. In particular the computed surface elevations from the 3D model are used to generate the curvature for the 2.5D case.

P3 L27: "Finally we discuss the importance of ...". I do not understand this sentence. "Certain authors"? Meaning the authors of this paper? Or Reeh?

P3 L31: "Similar synthetic geometry". Similar to what?

Sec. 2.1.: It would be useful to specify, as did Gillet-Chaulet and Hindmarsh (2011), that the edges of the domain do not correspond to a physical boundary. Indeed, the authors could draw a considerable amount of inspiration from that paper on how to describe the setup of the model experiments. To understand what was being done in this paper I had to read that paper, and most readers would appreciate eliminating the intermediate step.

Sec. 2.1.: Please distinguish between a dome (what is ostensibly being modeled) and a cylinder (the shape of the mesh).

Sec. 2.3.1: If the authors are unwilling to state the Stokes' equations completely, then it might be best to not state them at all, and just give a reference to one of Elmer/Ice's numerous model description papers. Otherwise, there are many missing definitions (e.g. $A(T)$, $\dot{\epsilon}$, etc.).

P4 L8: The strain rate tensor (written in the paper as $\epsilon$) needs a dot over it.

P5 L8: Calling $a$ a function is confusing in this context, because it's a constant. Just call it the accumulation rate.

Eqs. 9/10: Trigonometric functions are usually typeset upright, rather than in italics.

P5 L20: $L$ is not used, so why is it defined here?

P6 L2: It would be useful to use different coordinate symbols for the global Cartesian system that the 3D model uses versus the local system used by the 2.5D model.

P6 L13: Asserting that an assumption is reasonable requires a reference.

P6 Sec. 3.1: This section needs some clarification. Is it the ridge which runs along $y = 0$ or the centerline of the 2.5D model coordinate system?

P6 L24: If a flow tube diverges, then the tube surface area gets *larger*, and velocities are reduced because an equal flux moves through a larger area.

P7 L20: Why neglect transverse shear stresses (i.e. $\sigma_{xy}$)? Elmer can solve Stokes' equations, so technically it shouldn't be that difficult to include them here. There are plenty of width parameterizations out there (see, the works of Vanderveen on fjord wall drag, for example).

P7 L20: Please provide a page number for Jaeger (1969). Also, consider citing Hvidberg (1996) instead, as this published article is much more readily available than a PhD thesis. Also, according to Hvidberg (1996), this result is derived for the axisymmetric case. Can the authors show that it remains valid in the case where this assumption is violated (i.e. $\alpha > 2$). In any case, these equations need considerably more explanation.

P8 L3: Do the authors mean "imposed *horizontal* velocity profile"?

P8 L12: I like Elmer too, but its efficiency isn't really relevant here.

P8 Sec. 3.4: This section needs to be expanded greatly, and could do with some illustrative figures. After reading this, I really still have no idea where $R$ comes from, and why changing the size of the sample changes this. If the surface contours are well approximated by polynomials, I fail to see why a small window shouldn't perform equally to a large window. This is really the crux of the method and a discussion of it

takes up a bulk of the results, yet it is given only a few sentences in the methods. How can a reader assess the validity of the results without knowing what the authors did?

P9 L3: Calling the case with the closed-form $R = x$ "2D" rather than 2.5D is confusing. The extra half-dimension comes from the fact that width variations are being parameterized, and that's still the case when $R$ is known exactly. Much better would be to call these runs "2.5D with analytic $R$" or something like that.

Sec. 4.1.2: I wonder if using a different method to compute widths would be less error-prone. For example, since we're already assuming that the flow follows the surface gradient, why not try just finding two flowlines and computing the distance between them? That eliminates the need to compute a second derivative (usually an error prone activity). I guess if the rationale behind the way that curvature is computed were more fully explained, the answer to this question might be obvious.

Sec. 4.1.2: A bit of specificity beyond "an error range of 0-10%" is in order here. How about just reporting standard error?

Fig. 4,5,6,8: All need a legend.

P10 L7: I don't understand what this paragraph is saying, nor do I find any clues in Fig. 6. What is the significance of a concave surface slope?

P10 L19: "the error made with this more complete model seems now to be small enough to be used for dating purposes." What evidence presented herein supports this point? To make such a claim, the authors need to establish an error threshold (*a priori*) which must be met in order to claim that their method is accurate, and then go about showing quantitatively that the model performs up to this standard. At no point do I see any objective metrics for model performance in this regard.

P10 L25: What is the error here, and in what way is it consistent with Hvidberg (1997b)? Am I to take away that the 2.5D model does a better job perpendicular to the dome?

P10 L31: I am skeptical of the authors' hypothesis that there is significantly different

flow directions at different points in the ice column, primarily because this would lead to symmetry breaking that does not occur in any models that I know of. It would be easy to test this idea, since evidently the authors have the full 3D model output in hand. I don't really know what's causing the strange non-physical vertical inversion of the horizontal velocity profile, but I suspect it has to do with the neglect of transverse shear stresses or vertical resistive stresses.

P11 L7: I am not familiar with the results from Hvidberg (2002). It would be helpful if the authors restated them.

Sec 4.4: This section is a non-sequitur. What is the "mass-only conservation model"? Are the authors referring to a calculation of balance velocity? If so, there are numerical considerations and different boundary conditions that the authors do not state. Unless the authors make a considerable effort to define what the model results that they are referring to actually are, this section should be removed.

Conclusions: This sections seems to be an afterthought; it is too short, ambiguous, and the conclusions stated herein are not clearly supported by the text.

Appendix: with respect to typesetting, the dot used to indicate scalar multiplication is not necessary.

---

## Referee Comment (RC2) · Anonymous Referee #2 · 16 Mar 2016

This manuscript presents a study the performance of a 2.5D model versus a full 3D model and its applicability in the vicinity of a dome. Ice flow is complex and boundary conditions are not easily parameterized and well constrained by observations. Ice flow is described by a set of thermo-mechanically coupled non-linear differential equations and the numerical solution of these equations is computationally very demanding. Simulations of ice flow are often done on simplified systems, and a commonly used approximation is to reduce the 3D set of equations to a 2.5 flowline version. Investigating the applicability and performance of these models is therefore an interesting contribution to the community.

It is important to note that this test has not been done before. One reason is that only recently complex models that solves the full set of stress balance equations have become available to do the test. These models are still so computationally demanding,

however, that simplifications of the equations are required for many purposes.

The study is focused and well-structured. The model is presented, both the continuum mechanically based set of equations and boundary conditions as well as the numerical implementation. The set of equations are presented without further references and arguments for the choices of model parameters. It is clear that the model is run for Antarctic conditions (temperature conditions), but this is not mentioned. A little information on the choices of model parameters and possible effects would clarify (from the simplified temperature, and the chosen temperature regime).

More details on how R is determined from the DEM using a scanning window are needed, for example - how is the fit done, - explain that R is not constant within the window.... How R is determined is a critical parameter, for example the size of the scanning window, and the details of how this is done should be clearly described.

The effects when moving from 2.5D to 3D are complicated and result in surprising effects. It is surprising to see how the uncertainty of the radius of curvature for a small scanning window completely dominates in figure 5. It is very interesting to see the distribution of horizontal velocity fields for the non-isothermal case (figure 7). In 2.5D these variations may lead to spurious effects if used to model internal layers within the ice. A short paragraph should be included (introduction and/or conclusion) to mention this and thereby emphasize the significance of the results presented in the manuscript.

The manuscript only considers 2.5D flow along straight lines. Sometimes 2.5D models are being used along curved flowlines, and neglecting the curvature of the flowline would add further to the uncertainties. It would be difficult to say something general about curving flow lines, so I do not suggest further studies, but the problem with curving flow lines should be mentioned in the manuscript.

I do not understand the comparison presented in section 4.4. The mass-only conservation model is not explained in detail, and does not add further to the conclusions. I am also suspicious about the boundary effects near x= 15000 m. They are not discussed

but clearly influences the solution. I suggest that this section is removed.

There are several examples of incorrect use of English (e.g. order of words in a sentence), and I suggest that the manuscript is carefully worked through to clarify the text. The structure of the manuscript is well planned, and overall the manuscript appears clear and with a logical flow. The figures are clear and well presented.

To conclude, I find that the manuscript is relevant and provides a needed insights into the applicability of 2.5D models. The results can help clarify the performance and limitations of these models, which has not been systematically done before. The results also demonstrate that full stress solutions are needed near domes and divides to fully represent the flow. I recommend that the manuscript is published with minor changes mentioned above, as well as a thorough correction of the use of English in the text.

---

## Author Comment (AC1) · 27 Apr 2016

[12pt]article [utf8]inputenc [french]babel [T1]fontenc

helvet

float [parfill]parskip listings setspace multicol natbib,array calc geometry a4paper margin=2cm

[dvips]graphicx amsmath pstricks

[final]pdfpages    General Comments: For this paper to warrant publication, it would require its methods to be outlined in much greater detail than they currently are. The primary nuance of the work, mainly the computation of R from a surface DEM is given only a few lines of description. The metrics by which the authors judge quality are

primarily left unstated. Many modelling assumption (i.e. that surface elevations can act as a reasonable proxy for the direction of a Stokes'-based velocity field) are utilized but unjustifed. Suffice to say, this work would not be repeatable based upon the information given here. I would like to offer up an opinion on whether the results presented are significant to the field of glaciology, but lack a sufficient understanding of the quality of the results to do so.

Authors' answer: Several points needed to be specified to make the article more intelligible; in particular, we added information about the assumptions of the model (emphasize the fact that it is done for divides or center lines of drainage basins), and our method to process to the twin experiment between 3D and 2.5D.

The paper is rife with incorrect grammar, spelling, and otherwise improper English, to the extent that it partially inhibits understanding of the methods and results contained herein. As a reviewer, I don't really see it as my responsibility to correct for these issues, particularly when at least one of the co-authors is a native English speaker, and I will not provide comments on these points. As such, I would suggest that the authors employ or otherwise recruit a capable copy editor to polish the manuscript. It is presently not ready for publication for this reason, scientific merit aside.

Specific Comments: Abstract: the abstract introduces a significant amount of jargon, such as "scanning window", "reversed surface convexity", and "partly reversed velocity profile", which the reader cannot know the meaning of without reading the paper. This undercuts the purpose of the abstract.

Authors' answer: We explained the jargon, or removed it to make the sentences simpler. The english has been improved by a carreful re-reading from the english speaker of the team.

P1 L8: The relief in question is on the order of tens of meters, yet the authors suggest that the so-called "scanning window" should be on the order of kilometers. This directly contradicts this line.

Authors' answer: "relief" was erroneous. The sentence was modified to mention the radius of the surface curvature.

P1 L16: the equations in question are either Stokes' equations, or they are not; the "full-" modifier is unnecessary. Authors' answer: Modification done

P1 L21: Durand (2011) does not appear to actually address this issue, though it does use a flowline model. Also, what are the "regular flanks of an ice sheet"?

Authors' answer: Reference to Durand (2011) removed. "Ice-sheet" was changed in "ice cap". In Martin et al, 2006, the Roosevelt Island shows a sharp axis of symmetry, we made the sentence more specific.

P1 L22: "plane strain" does not seem to be used correctly here.

Authors' answer: Precision was added, but we don't agree and think that plane strain is the appropriate description of the state of strain we want to refer to.

P2 L1: I think I know what is meant by "a particular vertical surface", but this would be greatly clarified by addition of said surface to Fig 1.

Authors' answer: Fig.1 has been modified, showing the intersection between a vertical plane and the ice surface.

P2 L14: Why is the parenthetical "(column-flow model)" included here? It does not seem to serve a purpose and doesn't seem to be referenced later.

Authors' answer: 1-Since the 2.5D model is the only object of the paper, it's worth re-membering who developed this kind of model first, and who improved upon that. 2 - As we have some troubles with the velocity profiles computed with diverging geome-tries, column-flow models could still be useful (§4.3).

P2 L22: "In particular,..." I do not understand this sentence.

Authors' answer: We modified the sentence.

P2 L31: width of the flow tube?

Authors' answer: "Tube" added

P2 L31: A DEM gives the shape of the surface and its contour lines when velocity is available, as well.

Authors' answer: We changed the sentence, in the way that the DEM method is the only possible one when surface velocities are unknown.

P2 L34: Assuming that flow is always oriented along the surface gradient is a wrong assumption, and the differences between considering gradient-based curvature and velocity-based width goes far beyond issues of numerical accuracy. Have a look at any computation of balance velocity ever made; if flow is routed down-gradient then the surface velocity looks like a river network, which is silly. Longitudinal stresses matter for flow routing!

Authors' answer: The text is now more specific on the cases for which the model is built. In particular it should hold for flowlines along ice divides and center lines of drainage basins.

P2 L35: I don't know what "scanned" means, and as such I cannot assess whether the authors' statements about ambiguity in surface curvature have any merit. It would be better to hold off on elaborating upon it further until some basic definitions have been stated.

Authors' answer: We now give a basic description of the DEM method here.

P3 L10: On "diverging geometries": flow fields can diverge, because they are vector fields. Scalars cannot diverge, and the fields representing geometry are certainly scalars.

Authors' answer: Changed for "diverging flows"

P3 L17: What "1D parameter" are we talking about here? Where is this question

returned to in the rest of the text?

Authors' answer: The 1D parameter is R(x), which is explicitely mentionned now. We added a more specific answer to the question in the conclusion.

P3 L25: Please state more clearly that the 3D model is used to construct a set of surface geometries based on different choices of lateral boundary conditions. This 3D model output is then taken as input to the 2.5D model. In particular the computed surface elevations from the 3D model are used to generate the curvature for the 2.5D case.

Authors' answer: We added the suggested precisions.

P3 L27: "Finally we discuss the importance of ...". I do not understand this sentence. "Certain authors"? Meaning the authors of this paper? Or Reeh?

Authors' answer: Sentence deleted, since this particular experiment has been suggested to be removed.

P3 L31: "Similar synthetic geometry". Similar to what?

Authors' answer: "Similar" was erroneous

Sec. 2.1.: It would be useful to specify, as did Gillet-Chaulet and Hindmarsh (2011), that the edges of the domain do not correspond to a physical boundary. Indeed, the authors could draw a considerable amount of inspiration from that paper on how to describe the setup of the model experiments. To understand what was being done in this paper I had to read that paper, and most readers would appreciate eliminating the intermediate step.

Authors' answer: Additional explanations are provided, specifying the virtuality of the boundary and the way the BC is constructed.

Sec. 2.1.: Please distinguish between a dome (what is ostensibly being modeled) and a cylinder (the shape of the mesh).

Authors' answer: "cylinder" now replaces "dome"

Sec. 2.3.1: If the authors are unwilling to state the Stokes' equations completely, then it might be best to not state them at all, and just give a reference to one of Elmer/Ice's numerous model description papers. Otherwise, there are many missing definitions (e.g. $A(T), \epsilon$, etc.).

Authors' answer: We now specify a reference concerning the temperature dependence of the fluidity. The strain rate component are defined at the begining of the subsection.

P4 L8: The strain rate tensor (written in the paper as $\epsilon$) needs a dot over it.

Authors' answer: dot added

P5 L8: Calling a a function is confusing in this context, because it's a constant. Just call it the accumulation rate.

Authors' answer: "rate" replaces "function"

Eqs. 9/10: Trigonometric functions are usually typeset upright, rather than in italics.

Authors' answer: upright done

P5 L20: L is not used, so why is it defined here?

Authors' answer: We now specify that x=L is the downstream position, just for the reader to understand why the subscript L appears.

P6 L2: It would be useful to use different coordinate symbols for the global Cartesian system that the 3D model uses versus the local system used by the 2.5D model.

Authors' answer: Since we never need a curvilinear system in our case, defining additional notations could be confusing. We mention the curvilinear system because the model was primarily designed with these coordinates.

P6 L13: Asserting that an assumption is reasonable requires a reference.

[Figure]

Authors' answer: Reference added (Reeh, 1988)

P6 Sec. 3.1: This section needs some clarification. Is it the ridge which runs along y = 0 or the centerline of the 2.5D model coordinate system?

Authors' answer: Specification is given. y=0 and x=0 are taken on the 3D dome, and the 2.5D is run along one of these direction.

P6 L24: If a flow tube diverges, then the tube surface area gets larger, and velocities are reduced because an equal flux moves through a larger area.

Authors' answer: For a given output width, the more divergent the flow, the narrower the tube. For example : if there is no divergence, all the Fig. 1 would be gray, and for linear divergence (axisymmetry), half of it would be gray. For higher divergence, the tube would be even narrower.

P7 L20: Why neglect transverse shear stresses (i.e. $\sigma_{xy}$ )? Elmer can solve Stokes' equations, so technically it shouldn't be that difficult to include them here. There are plenty of width parameterizations out there (see, the works of Vanderveen on fjord wall drag, for example).

Authors' answer: We wanted to use a 2.5D model because the geometry of our future area of interest in Antarctica well corresponds to the assumptions of Reeh (1988). That's why we are putting ourselves in its direct following. In this frame, including transverse stresses would not allow to get rid of the y-coordinate. Concerning this questioning, the additional explanations of the model of Reeh (1988) in the introduction can help as well.

P7 L20: Please provide a page number for Jaeger (1969). Also, consider citing Hvidberg (1996) instead, as this published article is much more readily available than a PhD thesis. Also, according to Hvidberg (1996), this result is derived for the axisymmetric case. Can the authors show that it remains valid in the case where this assumption is violated (i.e. $\alpha > 2$). In any case, these equations need considerably more explanation.

Authors' answer: The model can be considered as a generalized axisymmetrical model (as you can see in Hvidberg et al., Ice flow between the Greenland Ice Core Project and Greenland Ice Sheet Project 2 boreholes in central Greenland, 1997). The paper of Hvidberg 1996 only deals with axisymmetry since she specifies that R=x, but she gives the general version of the eqations anyway. More information about the equations are now provided.

P8 L3: Do the authors mean "imposed horizontal velocity profile"?

Authors' answer: "Horizontal" now replaces "Vertical"

P8 L12: I like Elmer too, but its efficiency isn't really relevant here.

Authors' answer: Removed

P8 Sec. 3.4: This section needs to be expanded greatly, and could do with some illustrative figures. After reading this, I really still have no idea where R comes from, and why changing the size of the sample changes this. If the surface contours are well approximated by polynomials, I fail to see why a small window shouldn't perform equally to a large window. This is really the crux of the method and a discussion of it takes up a bulk of the results, yet it is given only a few sentences in the methods. How can a reader assess the validity of the results without knowing what the authors did?

Authors' answer: We give more explanations, in particular concerning the difference between large and small windows. In fact, the question is what is the typical distance which influences the local ice flow, and how we could account for it.

P9 L3: Calling the case with the closed-form R = x "2D" rather than 2.5D is confusing. The extra half-dimension comes from the fact that width variations are being parameterized, and that's still the case when R is known exactly. Much better would be to call these runs "2.5D with analytic R" or something like that.

Authors' answer: We changed the formulation for "2.5D"

[Figure]

Sec. 4.1.2: I wonder if using a different method to compute widths would be less error-prone. For example, since we're already assuming that the flow follows the surface gradient, why not try just finding two flowlines and computing the distance between them? That eliminates the need to compute a second derivative (usually an error prone activity). I guess if the rationale behind the way that curvature is computed were more fully explained, the answer to this question might be obvious.

Authors' answer: Even if we determined the width of the flow tube by tracking particles, the equations are anyway parameterized with $R$. As given by the equation 12, we would need to compute the derivative of the width. Furthermore, the method to track particles would not be efficient to evaluate the divergence on the top of the ridge: it would be impossible to distinguish between the particles, since they would be inside the same pixel on part of the trajectory (and even 100 % of the trajectory if the tube is much diverging).

Sec. 4.1.2: A bit of specificity beyond "an error range of 0-10%" is in order here. How about just reporting standard error?

Authors' answer: We now indicate the root mean square error (standard error could lead to compensation between negative and positive terms, and the final value could be deceptive.)

Fig. 4,5,6,8: All need a legend.

Authors' answer: The legends were added.

P10 L7: I don't understand what this paragraph is saying, nor do I find any clues in Fig. 6. What is the significance of a concave surface slope?

Authors' answer: We describe more specifically the shape of the surface. The problem that appears is numerical, and comes from the balance between the terms of the mass conservation equation when close to the singularity.

P10 L19: "the error made with this more complete model seems now to be small

enough to be used for dating purposes." What evidence presented herein supports this point? To make such a claim, the authors need to establish an error threshold (a priori) which must be met in order to claim that their method is accurate, and then go about showing quantitatively that the model performs up to this standard. At no point do I see any objective metrics for model performance in this regard.

Authors' answer: The text now mention the tracking of the ice particles, which is linked to the velocity field (so it is the same idea, but not mentionning a totally different subject).

P10 L25: What is the error here, and in what way is it consistent with Hvidberg (1997b)? Am I to take away that the 2.5D model does a better job perpendicular to the dome?

Authors' answer: We give the RMSE for these cases. The 2.5D model meets difficulties to handle high divergence, so it is logical that the results are better perpendicular to the sharpest ridge. The reference to Hvidberg et al. (1997b) was simply indicative, and the error in u is in fact of the same order of magnitude than the error in R. It would be quite heavy to properly show it here, so we removed the sentence.

P10 L31: I am skeptical of the authors' hypothesis that there is significantly different flow directions at different points in the ice column, primarily because this would lead to symmetry breaking that does not occur in any models that I know of. It would be easy to test this idea, since evidently the authors have the full 3D model output in hand. I don't really know what's causing the strange non-physical vertical inversion of the horizontal velocity profile, but I suspect it has to do with the neglect of transverse shear stresses or vertical resistive stresses.

Authors' answer: The only parameter that changes between the two experiments is the temperature. If the transverse shear stress or vertical resistive stress were involved, they would occur for the isothermal case as well. The enlargement of the tube, consequence of the different orientation of the flow along the vertical, comes from the 3D model. It is possible that, this phenomenon being very particular and problematic only

in the case of 2.5D model, it has received little attention.

P11 L7: I am not familiar with the results from Hvidberg (2002). It would be helpful if the authors restated them.

Authors' answer: More specific informations on Hvidberg et al. (2002) have been added.

Sec 4.4: This section is a non-sequitur. What is the "mass-only conservation model"? Are the authors referring to a calculation of balance velocity? If so, there are numerical considerations and different boundary conditions that the authors do not state. Unless the authors make a considerable effort to define what the model results that they are referring to actually are, this section should be removed.

Authors' answer: This section is removed.

Conclusions: This sections seems to be an afterthought; it is too short, ambiguous, and the conclusions stated herein are not clearly supported by the text.

Authors' answer: Two additional paragraphs have been added that steps back and takes stock.

Appendix: with respect to typesetting, the dot used to indicate scalar multiplication is not necessary.

Authors' answer: dot deleted

---

## Author Comment (AC2) · 27 Apr 2016

[12pt]article [utf8]inputenc [french]babel [T1]fontenc

helvet

float [parfill]parskip listings setspace multicol natbib,array calc geometry a4paper margin=2cm

[dvips]graphicx amsmath pstricks

[final]pdfpages   This manuscript presents a study the performance of a 2.5D model versus a full 3D model and its applicability in the vicinity of a dome. Ice flow is complex and boundary conditions are not easily parameterized and well constrained by observations. Ice flow is described by a set of thermo-mechanically coupled non-linear

differential equations and the numerical solution of these equations is computationally very demanding. Simulations of ice flow are often done on simplified systems, and a commonly used approximation is to reduce the 3D set of equations to a 2.5 flowline version. Investigating the applicability and performance of these models is therefore an interesting contribution to the community.

It is important to note that this test has not been done before. One reason is that only recently complex models that solves the full set of stress balance equations have become available to do the test. These models are still so computationally demanding, however, that simplifications of the equations are required for many purposes.

The study is focused and well-structured. The model is presented, both the continuum mechanically based set of equations and boundary conditions as well as the numerical implementation. The set of equations are presented without further references and arguments for the choices of model parameters. It is clear that the model is run for Antarctic conditions (temperature conditions), but this is not mentioned. A little information on the choices of model parameters and possible effects would clarify (from the simplified temperature, and the chosen temperature regime).

Authors' answer: We added some information about our global frame (our final goal is to work on a small dome in Antarctica), and we are more specific on the effects of temperature on viscosity.

More details on how R is determined from the DEM using a scanning window are needed, for example - how is the fit done, - explain that R is not constant within the window. . .. How R is determined is a critical parameter, for example the size of the scanning window, and the details of how this is done should be clearly described.

Authors' answer: More information is given concerning the determination of $R$ during our twin experiment, especially what we expect from a small or a large window. In fact, the question is what is the typical distance which influences the local ice flow.

The effects when moving from 2.5D to 3D are complicated and result in surprising effects. It is surprising to see how the uncertainty of the radius of curvature for a small scanning window completely dominates in figure 5. It is very interesting to see the distribution of horizontal velocity fields for the non-isothermal case (figure 7). In 2.5D these variations may lead to spurious effects if used to model internal layers within the ice. A short paragraph should be included (introduction and/or conclusion) to mention this and thereby emphasize the significance of the results presented in the manuscript.

Authors' answer: The issue of modelling the internal layers, which is indeed a possible goal of using such a model, is now addressed in the conclusion.

The manuscript only considers 2.5D flow along straight lines. Sometimes 2.5D models are being used along curved flowlines, and neglecting the curvature of the flowline would add further to the uncertainties. It would be difficult to say something general about curving flow lines, so I do not suggest further studies, but the problem with curving flow lines should be mentioned in the manuscript.

Authors' answer: This issue is now mentionned in the conclusion as well.

I do not understand the comparison presented in section 4.4. The mass-only conservation model is not explained in detail, and does not add further to the conclusions. I am also suspicious about the boundary effects near x= 15000 m. They are not discussed but clearly influences the solution. I suggest that this section is removed.

Authors' answer: As both referees suggested to remove this section, we removed it.

There are several examples of incorrect use of English (e.g. order of words in a sentence), and I suggest that the manuscript is carefully worked through to clarify the text. The structure of the manuscript is well planned, and overall the manuscript appears clear and with a logical flow. The figures are clear and well presented.

To conclude, I find that the manuscript is relevant and provides a needed insights into the applicability of 2.5D models. The results can help clarify the performance and

limitations of these models, which has not been systematically done before. The results also demonstrate that full stress solutions are needed near domes and divides to fully represent the flow. I recommend that the manuscript is published with minor changes mentioned above, as well as a thorough correction of the use of English in the text.

---

## Referee Report (RR1)

**Review of *Performance and applicability of a 2.5D ice-flow model in the vicinity of a dome**

Doug Brinkerhoff

May 20, 2016

This is my second review of *Performance and applicability of a 2.5D ice-flow model in the vicinity of a dome*. I am glad to see that the authors addressed many of my and another reviewers' comments, and the paper is much improved, both in content and syntax. However, I maintain two significant issues with both the approach and presentation of the work.

1. I remain uncomfortable with the authors' neglect of lateral shear stresses. In their response to my initial comment on this issue (P7 L20 in the first manuscript version), the authors claim that they cannot maintain the influence of lateral shear stresses while operating in a topologically 2D computational framework. This is patently false: See the myriad papers on lateral drag in fjord flowline models as used by authors such as Van der Veen, Enderlin, Nick, and many others. The key is integration over the transverse coordinate, followed by the assumption that values in the longitudinal coordinate are averaged over the transverse one.

   I recognize that the authors' line of reasoning follows directly from Reeh (1988), who claims that lateral shear stresses are zero at ice divides and at basin centerlines (defined as a local velocity maximum), and that this is a justification for neglecting the transverse stress term in a flowline model. However, this seems to contradict the possiblity of lateral drag. The reason that this is false is because it is really the transverse derivative of lateral shear stress which balances driving stress, which is generally nonzero. Alternatively, the effect of such terms can be seen by (once again) integrating across the transverse coordinate. One immediately sees that there are lateral drag terms which emerge. I don't necessarily think that neglecting this term is a fatal flaw; I do think that this assumption ought to be stated, justified, and discussed in terms of the potential errors that it could induce in the model results.

2. The method for computing curvature doesn't make sense to me (though perhaps this is just my own ignorance). The definition of radius of curvature is the radius of the circle that locally approximates a curve. If you have surface elevation contours for each point at which you need $R(x)$, why not just fit a circle to the points near your flowline? I'm particularly nervous about the use of a bivariate interpolant: The object that you need to take the curvature of is topologically 1D (a surface contour). What is the physical

meaning of the curvature of a bivariate function? The radius of a local spherical approximation of the surface? If the latter, then that's not right. Furthermore, the last sentence of the added paragraph states that $R(x)$ is being taken as the *inverse* of the curvature of the contours, which is definitely wrong (presumably a typo, but I can't say for certain). I think a much more quantitative and rigorous description of exactly what was done to compute this radius of curvature is in order here; ultimately, I still don't understand the methods here, yet they remain the crux of the paper. If this work is based on existing methods, a reference might be nice too.

Specific Comments (Somewhat abridged until above major comments are addressed):

- P1 L20: Statement about computing time needs a citation or at least an explanation.

- P1 L24: When I said that the Durand paper wasn't applicable, I didn't mean that a citation was not needed.

- P1 L24: Maybe specify that you mean an ice cap which exhibits mirror symmetry. This isn't true for an axisymmetric one: just look at the difference in divide heights between the 2D and 3D EISMINT experiments.

- P5 L26: 'dome surface *area*'

- P6 L8: Actually globally Cartesian in the case of straight flowlines

- P8 Eq. 19: surface mass balance is traditionally typeset as $\dot{b}$ or $\dot{a}$ in glaciological literature, but that's just a notational choice.

- P8 L22: 'representative of a real ice sheet'. Citation needed.

- P9 4.1.1: Should this heading be 'analytical comparision' or something like that?

Finally some responses to a few of the rebuttals that the authors made towards my initial review.

- I asked why not use the RHS of Eq. 12 via particle tracking, rather than computing radius of curvature. The authors responded that this would be inefficient and that it would not work on a ridge. I still fail to see why this would be true. It seems to me that doing what I suggested would provide something very much like the authors' Figure 1, from which (a differentiable) $W(x)$ could be computed. At the very least, explicitly computing the flow tube would provide a boundary for the region over which $R(x)$ should be computed. I'm not trying to get the authors to change their methods here, mostly just curious at this point.

- I suggested using a more precise metric for error quantification, such as standard error. I would be happy with the authors' choice of using RMSE, however that doesn't seem to be what they are reporting. RMSE has units, and the error here is reported in terms of percentages. I would like a more explicit statement of what is actually being computed in terms of error (namely, how is RMSE being normalized?).